# The Effect of Bi_2_O_3_ and Fe_2_O_3_ Impurity Phases in BiFeO_3_ Perovskite Materials on Some Electrical Properties in the Low-Frequency Field

**DOI:** 10.3390/ma15144764

**Published:** 2022-07-07

**Authors:** Cristian Casut, Iosif Malaescu, Catalin Nicolae Marin, Marinela Miclau

**Affiliations:** 1Physics Faculty, West University of Timisoara, V. Pârvan Ave., No. 4, 300223 Timisoara, Romania; cristian.casut95@e-uvt.ro (C.C.); iosif.malaescu@e-uvt.ro (I.M.); 2National Institute for Research and Development in Electrochemistry and Condensed Matter, Plautius Andronescu Str., No. 1, 300224 Timisoara, Romania; marinela.miclau@gmail.com; 3Institute for Advanced Environmental Research, West University of Timisoara (ICAM-WUT), Oituz Str., No. 4, 300086 Timisoara, Romania

**Keywords:** bismuth ferrite, electrical conductivity, complex dielectric permittivity, VRH model, thermal activation energy of conduction, Bi_2_O_3_, Fe_2_O_3_, BFO

## Abstract

Pure bismuth ferrite (BFO) and BFO with impurity phases (Bi_2_O_3_ or Fe_2_O_3_) were synthesized by the hydrothermal method. Complex dielectric permittivity (ε) and electrical conductivity (σ) were determined by complex impedance measurements at different frequencies (200 Hz–2 MHz) and temperatures (25–290) °C. The conductivity spectrum of samples, σ(f), complies with Jonscher’s universal law and the presence of impurity phases leads to a decrease in the static conductivity (σ_DC_); this result is correlated with the increased thermal activation energy of the conduction in impure samples compared to the pure BFO sample. The conduction mechanism in BFO and the effect of impurity phases on σ and ε were analyzed considering the variable range hopping model (VRH). Based on the VRH model, the hopping length (R_h_), hopping energy (W_h_) and the density of states at the Fermi level (N(E_F_)) were determined for the first time, for these samples. In addition, from ε(T) dependence, a transition in the electronic structure of samples from a semiconductor-like to a conductor-like behavior was highlighted around 465–490 K for all samples. The results obtained are useful to explain the conduction mechanisms from samples of BFO type, offering the possibility to develop a great variety of electrical devices with novel functions.

## 1. Introduction

The bismuth ferrite (BiFeO_3_), BFO as it will henceforth be referred to, is a compound with a perovskite structure that has received considerable attention because of its technical applications [1,2], being one of the most promising multiferroic materials [1,3]. In general, the bismuth ferrite is metastable in air, with spots of impurities that appear below the melting temperature [4] being optically visible. The most important properties of BFO are its Curie (T_C_ = 1103 K) and Néel temperatures (T_N_ = 643 K) that allow this compound to keep its properties at extreme temperatures. The phase of the compound BFO at room temperature is rhombohedral (R3c) [5,6], having a rhombohedral angle with a value between 89.3° and 89.48° [7,8]. BFO has been intensively studied because of its potential applications in many domains due to both ferroelectric and ferromagnetic properties [9].

Various synthesis methods such as sol–gel [10], hydrothermal [11], microwave hydrothermal [12] and microemulsion technique [13] have been employed in an attempt to obtain nanosize and monocrystalline BFO particles. Particle morphology also plays an important role in determining BFO ferroelectric performance [9]. However, it is important to note that, in most cases, it is difficult to obtain BFO particles with nanoscale size, single crystalline phase and favorable morphology due in part to the secondary phase formation in the Bi_2_O_3_-Fe_2_O_3_ system [14]. Therefore, it is still a big challenge for scientists and engineers to achieve single crystalline BFO nanoparticles with favorable morphology.

In this paper, BFO polycrystalline materials are obtained by the hydrothermal method. The synthesized samples, both single-phase BFO and BFO with impurity phases (Bi_2_O_3_ or Fe_2_O_3_), were studied by Scanning Electron Microscopy (SEM) and X Ray Diffraction (XRD) analysis to study their morphology and structure.

More authors have determined some electrical and dielectric properties for materials such as BiFeO_3_ thin films [15,16,17], for BiFeO_3_ doped with rare earths [18] and for perovskite ferroelectrics (Ba, Sr)TiO_3_ type, in the microwave field [19]. Moreover, theoretical calculations allowed the determination of the band gap energy both for bulk and thin film BFO structures [20], the obtained values being 2.7 eV (bulk) and 2.5 eV (thin film), with the authors showing that the conduction band contributes most to these differences. The determination of the electrical conductivity (σ) and dielectric permittivity (ε) of the bismuth ferrite crystalline compounds and experimental studies regarding the low frequency and temperature dependence of σ and ε are few in the literature.

As a result, the purpose of our work is to study the conduction mechanisms of BFO samples and how the Bi_2_O_3_ and Fe_2_O_3_ impurity phases affect those mechanisms, taking into account Mott’s theoretical VRH model [21,22], with the possibility of developing new applications in the fields of electronics, memory devices and thermo-electric sensors, which could be electrically approached. Furthermore, the effect of the phase impurities in BFO material on the activation thermal energy of conduction and of the Mott parameters were studied. For this, experimental measurements of the electrical conductivity and of the complex dielectric permittivity over a wide temperature and frequency range were performed.

## 2. Materials and Methods

For the preparation of the BFO sample, the hydrothermal synthesis route was used. For this, the solutions of bismuth nitrate (Bi(NO_3_)_3_∙5H_2_O) and iron nitrate (Fe(NO_3_)_3_∙9H_2_O) in distilled water were mixed in stoichiometric proportions (1:1 mole ratio), resulting in 10 mL of solution. This resultant solution and 10 mL of 1 M NaOH solution were put together under vigorous stirring before being transferred to a Teflon line autoclave that was then closed and put into an oven at 200 °C for 12 h. After this procedure, the precipitate in the form of a brown powder was removed and washed with double distilled water before drying at 70 °C to obtain the final polycrystalline BFO powder. The sample obtained was a pure bismuth ferrite (BiFeO_3_), denoted as Sample S1 (BFO). Similarly, both a slight change in the quantities of bismuth nitrate and iron nitrate and washing the precipitate with distilled water removed from the autoclave leads to the obtaining of two other samples of bismuth ferrite but also containing impurities, denoted by sample S2 (BFO with Bi_2_O_3_ impurities phase) and sample S3 (BFO with Fe_2_O_3_ impurities phase).

To check the crystal structure of the obtained samples, X-ray powder diffraction was performed with an XRD PANalytical X’Pert PRO MPD Diffractometer (Almelo, The Netherlands) using Cu-K radiation operated at 40 kV and 30 mA over the 2θ range of 20°–60°.

The morphology, structures and stoichiometric composition of samples, as well as the size of the impurity phase, were investigated by Scanning Electron Microscopy (SEM/EDX Inspect S model, Eindhoven, The Netherlands).

For electric and dielectric measurements, all three samples of polycrystalline BFO powder were pressed into disks of the same sizes (6 mm diameter and approximately 1 mm thickness) after being mixed with a binder solution (5% PVA-polyvinyl alcohol). The disk was dried and then sintered in air at 100 °C and 700 °C, respectively. A layer of Ag was deposited on both polished surfaces of the disk, representing the two electrodes that were connected to an LCR-meter, with which the measurements of the complex impedance of investigated samples were performed.

The electrical conductivity and the complex dielectric permittivity of samples were determined by complex impedance measurements at different frequencies between 200 Hz and 2 MHz and at different temperatures from 25 to 290 °C, using an LCR meter TEGAM model 3550 (Geneva, OH, USA) and an electrical oven [23].

## 3. Results and Discussion

### 3.1. Structural and Morphological Properties

The X-ray diffraction pattern of BFO powders obtained by the hydrothermal method is presented in Figure 1. All diffraction peaks for sample 1 (Figure 1a) are indexed as BiFeO_3_ (JCPDS no. 01-072-2321) with a rhombohedral structure, and no formation of the impurity phases during the synthesis route was observed in sample S1. The formation of Bi_2_O_3_ and Fe_2_O_3_ impurity phases can be observed in sample S2 (Figure 1b) and sample S3 (Figure 1c), respectively. The amounts of the impurity phases were roughly quantified using the XRD data. The impurity contents are 13% Bi_2_O_3_ in S2 and 10% Fe_2_O_3_ in S3.

The crystallite size and the lattice strain were determined for each sample in accordance with the Williamson-Hall (W-H) equation [24]:(1)$$\beta \cos\theta =\frac{0.9\lambda }{D}+4\varepsilon \sin\theta$$

In Equation (1), λ is the wavelength of X-ray radiation, β is the full width at half maximum (FWHM) and θ is the diffraction angle of the diffraction peaks. D is the effective crystallite size with lattice strain, and ε is the effective value of the lattice strain. βcosθ was plotted against 4sinθ and, after linear fitting, the intercept gives the value of D, and the slope gives the value of ε. Figure 2 presents the W-H plot of all samples. The positive strain of all samples highlighted that the system is under tensile strain. According to Table 1, the presence of Bi_2_O_3_ and Fe_2_O_3_ influences the crystallization of BiFeO_3_ particles, as the crystallite size is reduced from 182.1 nm to 82.5 nm and 62.7 nm for Bi_2_O_3_ and Fe_2_O_3_ phase impurity, respectively, and at the same time decreasing the lattice strain.

The presence of the impurities on the BFO microstructure is shown in the SEM images (Figure 3). Figure 3a indicates that the shape of the resulting BFO is a large-scale agglomeration based on truncated and strongly distorted octahedrons. SEM images of S2 and S3 (Figure 3b,c) show two different types of morphologies for the Bi_2_O_3_ and Fe_2_O_3_ impurities. In Figure 3b, we can observe that on the BiFeO_3_ morphologies, the impurity phase formed as needle-like crystals. In Figure 3c, smaller nanoparticles of Fe_2_O_3_, with a size of 1–2 nanometers, are on the surface of BiFeO_3_.

### 3.2. Electrical Conductivity Analysis

#### 3.2.1. DC-Conductivity

The bismuth ferrite investigated sample, in the form of a disk, was connected to the LCR-meter (TEGAM model 3550) and placed in a thermally isolated oven, heated by an electrical resistor; the temperature (*T*) of the sample was registered with a thermocouple [23]. In this way, for each sample, Z′ and Z″ components of the complex impedance at different temperatures and frequencies were determined. Using Z′ and Z″, the electrical conductivity, σ, was calculated with Equation (2):(2)$$\sigma =\frac{Z^{\prime }}{Z^{\prime 2}+Z^{\prime \prime 2}}\cdot \frac{d}{A}$$

In Equation (2), d and A are the length and cross-sectional area of the sample, respectively.

Figure 4 shows the frequency dependence of the total conductivity (σ) of the investigated samples at room temperature.

The frequency dependence of conductivity has been explained by Jonscher [25] using Equation (3). In this equation, the static component σ_DC_(T) does not depend on frequency, whilst high-frequency component σ_AC_(ω, T) follows Jonscher’s universal law and is dependent both on frequency and temperature [25]. This component is correlated with the dielectric relaxation processes determined by the localized electric charge carriers [21,25], being given by the Equation (3):(3)$$\sigma =\sigma_{DC}(T)+\sigma_{AC}(\omega ,T)$$
(4)$$\sigma_{AC}(\omega ,T)=A_{0}(T){\left(2\pi f\right)}^{n(T)}$$

In Equations (3) and (4), ω = 2πf is the angular frequency, A_0_(T) is a temperature-dependent parameter, and the exponent *n* is dimensionless and temperature-dependent (n<1). As can be observed from Figure 4, the two regions of the conductivity spectrum are well defined: the static component (σ_DC_) up to frequency of 1–2 kHz and the dynamic component (σ_AC_) above 3 kHz, which increases rapidly with the frequency. From Figure 4, the σ_DC_ values of the static conductivity of the investigated samples were determined, obtaining the following values: $\sigma_{DC}=6.30\times 10^{-5}\;S/m$(for sample S1), $\sigma_{DC}=3.36\times 10^{-5}\;S/m$ (for sample S2) and $\sigma_{DC}=1.28\times 10^{-5}\;S/m$ (for sample S3). The results show that the presence of Bi_2_O_3_ or Fe_2_O_3_ impurity phases in the pure BFO sample leads to a decrease in the static conductivity of both S2 and S3 samples.

In the dispersion region of Figure 4, corresponding to high frequencies (f > 200 kHz), the high-frequency conductivity (σ_AC,_) follows a law according to Equation (4). Taking the logarithm of the equation, it results:(5)$$\ln\sigma_{AC}=\ln A_{0}+n(T)\ln(2\pi f)$$
which shows a linear dependence between lnσ_AC_ and lnω. Figure 5 shows the experimental dependence, ln(σ_AC_)(ln (2πf)), in the high-frequency range (f > 200 kHz) at room temperature for the investigated samples.

By fitting the experimental dependence, *ln*(*σ*)(*ln*(2*πf*)), from Figure 5 with a straight line, we can determine the exponent *n* corresponding to each sample based on Equation (5). The obtained values are: *n*(*S*1) = 0.297, *n*(*S*2) = 0.407 and *n*(*S*3) = 0.521. Appling the correlated barrier hopping (CBH) model [26], the n exponent can be written in the first approximation as follows:(6)$$n=1-\frac{6kT}{W_{m}}$$
where Wm is the maximum energy of the barrier (or band gap energy). Knowing the value of the exponent n and considering T = 300 K, from Equation (6) results: W_m_ = 0.222 eV for sample S1, W_m_ = 0.263 eV for sample S2, and W_m_ = 0.325 eV for sample S3. As a result, the presence of the impurities phases in the BFO sample leads to an increase in the W_m_ of samples S2 (BFO-Bi) and S3 (BFO-Fe), respectively, compared to the W_m_ of the sample S1 (BFO), with these values being correlated to the decrease in the values of the static conductivity (σ_DC_) of the samples S2 (BFO-Bi) and S3 (BFO-Fe) in relation to the sample S1 (BFO).

#### 3.2.2. Temperature Dependence of the DC-Conductivity

To investigate the temperature dependence of the σ_DC_ conductivity at a constant frequency from the static region, f = 0.5 kHz (see Figure 4), the sample connected to the LRC-meter was placed into the oven [23], and the components Z′ and Z″ at different temperatures between 25 °C and 290 °C were measured, and then, with Equation (2), σ_DC_ was computed. The temperature dependence of the conductivity (σ_DC_(T)) for the investigated samples is shown in Figure 6.

As can be seen from Figure 6, over the temperature range of 300 K–550 K, the σ_DC_ conductivity reaches a maximum at the temperature (T_max_), namely 475 K (for sample S1) and 467 K for the other two samples. T_max_ represents the transition temperature of the sample from a semiconductor type behavior to a conductive type behavior [19,22]. The increase in the temperature of the static conductivity σ_DC_(T), from 25 °C to T_max_, for all three samples suggests a thermally activated conduction process, in accordance with the VRH model of Mott and Davis [21,22]. By increasing the temperature, the hopping electrons will be thermally activated, which leads to an increase in the drift mobility of electrons and therefore an increase in σ_DC_ [19,27]. According to the VRH model, in the temperature range in which the samples are semiconductors (from 300 K to T_max_), the DC conductivity, σ_DC,_ is given by Equation (7):(7)$$\sigma_{DC}=\sigma_{0}\exp\left[-{\left(\frac{T_{0}}{T}\right)}^{1/4}\right]$$
where σ_0_ is the pre-exponential factor, and T_0_ represents the temperature characteristic coefficient being given by the relation [22]:(8)$$T_{0}=\frac{\lambda a^{3}}{kN(E_{F})}$$

In Equation (7), a ≈ 10^7^ cm^−1^ represents the degree of localization, *λ* is a non-dimensional constant with the value, λ ≈ 16.6, N(E_F_) is the density of states at the Fermi level and *k* is Boltzmann’s constant [22].

Taking the logarithm of Equation (7) results in:(9)$$\ln\sigma_{DC}=\ln\sigma_{0}-{\left(T_{0}\right)}^{1/4}\frac{1}{T^{1/4}}$$

Using the measured values of σ_DC_ over the temperature range where the samples are semiconductors (300 K–T_max_), the dependencies of lnσ_DC_ on 1/T^1/4^ were plotted for all three samples and presented in Figure 6. Fitting the experimental dependencies $\ln\sigma_{DC}(T^{-1/4})$ with a linear dependence, in the temperature range of 300–450 K, the temperature characteristic coefficient, T_0_, of each sample was determined.

Knowing the values of T_0_ from Equation (8), we computed the density of states at the Fermi level(N(E_F_)) for each investigated sample, with the obtained values being shown in Table 2.

Using the VRH model, the hopping length (R_h_) and the hopping energy (W_h_) were computed with Equations (10) and (11) [21,22], and the resulting values are presented in Table 2.
(10)$$R_{h}={\left(\frac{9}{8akTN(E_{F})}\right)}^{1/4}$$
(11)$$W_{h}=\frac{3}{4\pi R_{h}^{3}N(E_{F})}$$

As is shown in Table 2, for each sample, N(E_F_) does not depend on temperature but decreases due to the presence of the impurity phases in the bismuth ferrite. This decrease in N(E_F_) can be correlated with an increase in the hopping distance between the localized states of the charge carriers.

As can be observed from Table 2, R_h_ decreases and W_h_ increases with increasing temperature for all samples. In addition, the parameters R_h_ and W_h_ are higher for samples S2 and S3, which contain impurity phases of Bi_2_O_3_ or Fe_2_O_3_, than for the pure bismuth ferrite sample (sample S1), at any temperature (*T*) in the investigated temperature range, from 25 °C to T_max_.

In the VRH model, *σ_DC_*(*T*) can be written in another form [22]:(12)$$\sigma_{DC}=\sigma_{0}\exp\left(-\frac{B}{T^{1/4}}\right)$$
where
(13)$$B=\frac{4E_{A,cond}}{kT^{3/4}}$$
represents the slope of the experimental dependence, $\ln\sigma_{DC}(T^{-1/4})$_,_ from Figure 6 is the same with the slope (T_0_)^1/4^ of the same dependency resulting from Equation (4), *k* is the Boltzmann constant, and E_A,cond_ is the thermal activation energy of the electrical conduction corresponding to the samples. Taking into account the values determined for the slope (T_0_)^1/4^ (see inside Figure 7) and the expression (12) of slope B, by matching them, one obtains the following relation for E_A,cond_:(14)$$E_{A,cond}=\frac{k{\left(T_{0}\right)}^{1/4}T^{3/4}}{4}$$

In Figure 8, the dependencies of E_A,cond_ on temperature (*T*) are shown. As can be seen from Figure 8, E_A,cond_ varies linearly with the temperature for all samples so that the conduction mechanism from the samples can be explained by the hopping between the localized states of the charge carriers, according to the VRH model [21].

From Figure 8, it can also be observed that E_A,cond_ increases from 0.18 eV to 0.25 eV for sample S1, from 0.21 eV to 0.29 eV for sample S2 and from 0.23 eV to 0.32 eV (or sample S3, being in accordance with the values obtained by other authors [28,29], for similar samples. Therefore, the obtained results show that the thermal activation energy of the static conductivity of samples S2 (with Bi_2_O_3_ impurities) and S3 (with Fe_2_O_3_ impurities) is higher than that corresponding to the pure sample S1 (BFO), being in correlation with the σ_DC_ values obtained for these samples (see Figure 4).

### 3.3. Complex Dielectric Permittivity Analysis

#### 3.3.1. Frequency Dependence of the Complex Dielectric Permittivity

Using an LCR-meter TEGAM 3550 model, the real ($\varepsilon^{\prime }$) and imaginary ($\varepsilon^{\prime \prime }$) components of the complex dielectric permittivity, $\varepsilon^{*}(f)=\varepsilon^{\prime }(f)+i\varepsilon^{\prime \prime }(f)$_,_ at different frequencies between 200 Hz and 2 MHz at room temperature for all samples were measured, and the results are shown in Figure 9.

As can be seen from Figure 9, the component ε′ of the complex dielectric permittivity gradually decreases with frequency over the entire interval for all three samples. At the beginning of the frequency range (200 Hz), ε′ of sample S1 (BFO without impurities) has the highest value (ε′ = 3400), whilst ε′ decreases to 1750 for sample S2 (BFO-Bi_2_O_3_) and to 500 for sample S3 (BFO-Fe_2_O_3_); comparable values were obtained by other authors for similar samples [30]. The rapid decrease in the frequency of the dielectric constant ε′ (Figure 9) shows that, in the samples with high values of ε′, the dispersion is large as compared to the samples with small values of ε′ [31,32].

In addition, from Figure 9, it is observed that up to the frequency of around 0.5 kHz, the values of ε″ are higher than the values of ε′, which shows that in these samples the conduction losses are high. Above 0.5 kHz, the values of ε″ fall below the values of ε′ for all samples, indicating that the conduction losses decrease and the dielectric relaxation becomes prevalent.

According to Debye’s equations [33], the imaginary component ε″ is given by the relation:(15)$$\varepsilon^{\prime \prime }=\varepsilon_{cond}^{\prime \prime }+\varepsilon_{rel}^{\prime \prime }=\frac{\sigma_{DC}}{2\pi f\varepsilon_{0}}+\varepsilon_{rel}^{\prime \prime }$$
where $\varepsilon_{cond}^{\prime \prime }$ is the component due to electrical conduction, and $\varepsilon_{rel}^{\prime \prime }$ is the component due to dielectric relaxation. Possible dielectric relaxation processes in the samples can be investigated by determining the relaxation component of the permittivity ($\varepsilon_{rel}^{\prime \prime }$). For this, using the measured values of ε″ and σ_DC_, the relaxation component of the imaginary part of the complex dielectric permittivity of samples was determined from the relation 14. In Figure 10, the frequency dependence of the imaginary component of complex dielectric permittivity of samples due to dielectric relaxation ($\varepsilon_{rel}^{\prime \prime }$) is shown.

As can be seen from Figure 10, $\varepsilon_{rel}^{\prime \prime }$ shows a maximum for all samples: at a frequency f_max_ (S1) = 20 kHz for sample S1; f_max_ (S2) = 18.52 kHz for sample S2; and f_max_ (S3) = 9.85 kHz for sample S3, indicating the existence in these samples of an interfacial dielectric relaxation [34]. The corresponding relaxation time, τ, can be determined using the Debye equation, 2*πf_max_τ* = 1 [35], and the values obtained are: τ(S1) = 7.96 μs; τ(S2) = 8.60 μs; and τ(S3) = 16.16 μs, respectively.

#### 3.3.2. Temperature Dependence of the Complex Dielectric Permittivity

In order to investigate the temperature dependence of the dielectric permittivity of the samples, measurements of the real component, ε′, were performed at different temperatures between 25 °C and 280 °C and at a constant frequency (*f* = 0.5 kHz) from the DC region (see Figure 4), thus obtaining the temperature dependence(ε′(T)) for the three samples, as shown in Figure 11.

From Figure 11, it is observed a maximum of the real component, ε′, of dielectric permittivity for each sample at the temperature T_max,1_ = 488 K (for sample S1), T_max,2_ = 470 K for sample S2 and T_max,3_ = 467 K for sample S3. As we showed in Section 3.2.2, the samples are characterized by a hopping process of charge carriers, thermally activated, which determines their electrical conductivity. Moreover, the temperature dependence of σ_DC_ (see Figure 6) looks similar to the temperature dependence of ε′ (Figure 11), with maxima at comparable temperatures. The increase in the conductivity of samples due to the thermally activated hopping of charge carriers leads to the increase in the electric charge accumulated at the grain boundaries in the samples. This is equivalent to an increase in the electric dipoles associated with each grain and to the increase in the real component of the complex dielectric permittivity. Therefore, a similar behavior with a temperature of σ_DC_ and ε′ proves that the dielectric polarization in the investigated samples is determined by the interfacial mechanisms of the Maxwell–Wagner type [36]. At the same time, in addition to the polarization due to the electron hopping and interfacial polarization, an ionic polarization may occur in the investigated samples [27].

The values obtained by us for the temperature at which the ε′ is maximum (see Figure 10) are close to those observed by other authors [37,38,39] for composite ceramic samples of perovskite type located between 453 K and 628 K, with variations depending on the composition of the sample, particle size and frequency at which the dependence on temperature of *ε*′ was determined.

## 4. Conclusions

Single-phase bismuth ferrite (BFO) and BFO with impurity phases (of Bi_2_O_3_ and Fe_2_O_3_) were prepared using the hydrothermal method, and then X-ray diffraction and SEM analysis were used to examine the structural and morphological properties of the samples.

The frequency and temperature dependencies of electrical conductivity (σ) and of dielectric permittivity (ε) over the frequency range of 200 Hz–2 MHz and different temperatures from 25 °C to 290 °C were determined. The spectrum of conductivity (σ(f)) at constant temperature follows Jonscher’s universal law.

Based on the temperature dependence of σ_DC_, a transition of the samples from a semiconductor-like behavior to a conductor-like behavior was highlighted at temperatures between 465 and 490 K.

Using the temperature dependence of electrical conductivity at a constant value of frequency (*f* = 0.5 kHz) and based on the model VRH, the conduction mechanisms in samples and the effect of Bi_2_O_3_ and Fe_2_O_3_ impurity phases in the BFO sample on the static conductivity σ_DC_ were analyzed. The presence of impurity phases in BFO samples induces higher thermal activation energies of conduction, with this result being correlated to the decrease in the static conductivity (σ_DC_) in these samples compared to the pure BFO sample.

Based on the VRH model, the hopping length (R_h_) the hopping energy (W_h_) and the density of states at the Fermi level (N(E_F_)) were determined for the first time in the case of these samples. Both R_h_ and W_h_ increase at a constant temperature in the presence of impurity phases compared to R_h_ and W_h_ values from the pure BFO sample. Furthermore, the density of states at the Fermi level (N(E_F_)) decreases in the presence of the impurity phases compared to the N(E_F_) value from the pure BFO sample. This decrease in N(E_F_) was attributed to the increase in the hopping distance between the localized states of the charge carriers.

The obtained results show that at low frequencies (up to approximately 0.5 kHz), the component ε″ has values much higher than ε′, thus resulting that, up to these frequencies, the conduction losses in the samples are predominant with respect to those due to the dielectric relaxation. Considering the values obtained for the static conductivity (σ_DC_) of the samples and eliminating the losses due to electrical conduction, a dielectric relaxation process was revealed, which was attributed to the interfacial polarization.

These results are very useful to explain the conduction mechanisms in BFO samples, both single-phase and with impurity phases (of Bi_2_O_3_ and Fe_2_O_3_), offering the possibility to develop a large variety of electronic and electrical devices with new functions.

## Figures and Tables

**Figure 1 materials-15-04764-f001:**
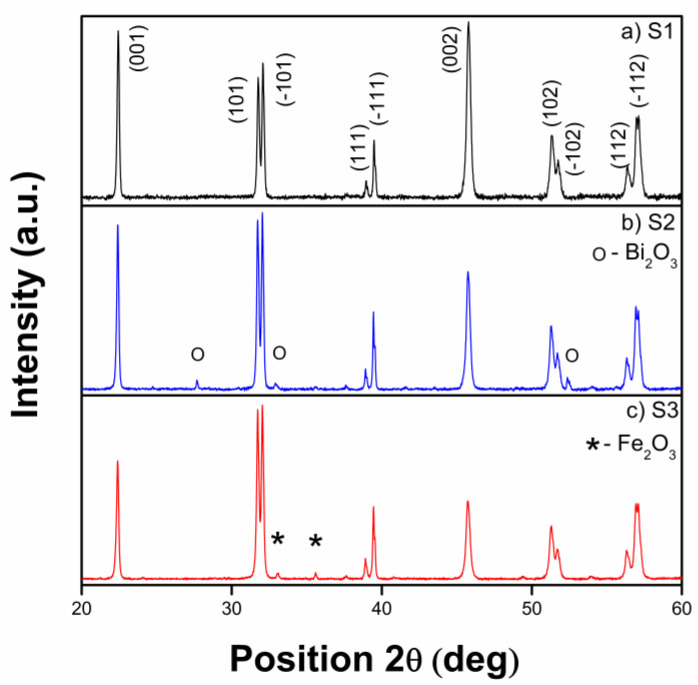
X-ray diffraction pattern for pure BFO (**a**), BFO with Bi_2_O_3_ impurity phase (**b**) and BFO with Fe_2_O_3_ impurity phase (**c**).

**Figure 2 materials-15-04764-f002:**
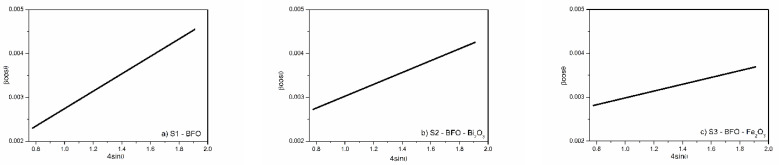
W-H plot of (**a**) BFO, (**b**) BFO with Bi_2_O_3_ and (**c**) BFO with Fe_2_O_3_.

**Figure 3 materials-15-04764-f003:**
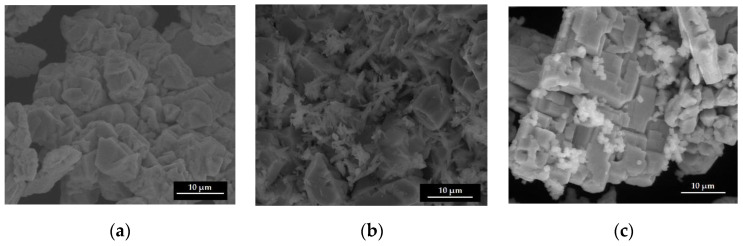
SEM images of BFO (**a**), BFO with Bi_2_O_3_ impurity phase (**b**) and BFO with Fe_2_O_3_ impurity phase (**c**).

**Figure 4 materials-15-04764-f004:**
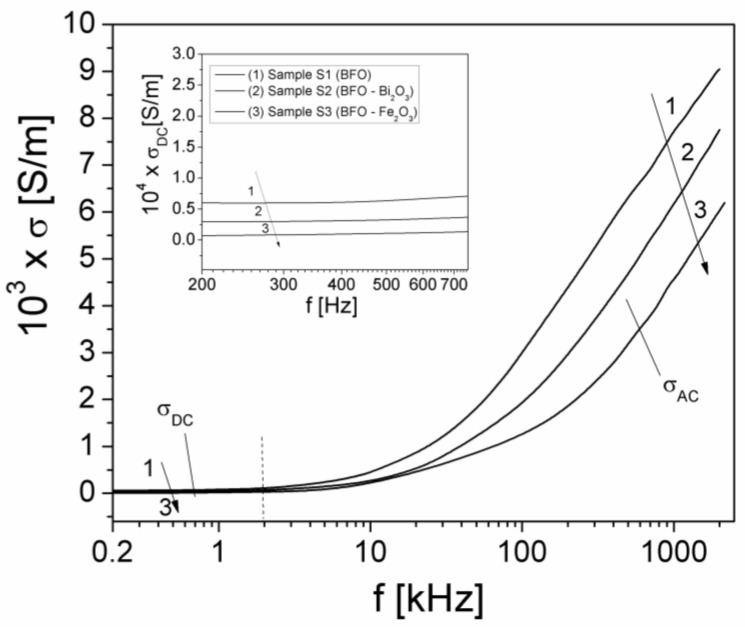
The frequency dependence of the conductivity, *σ*, of samples at room temperature (the inset shows the σ_DC_(f) dependence in a frequency range from 200 to 750 Hz when σ can be approximated with σ_DC_).

**Figure 5 materials-15-04764-f005:**
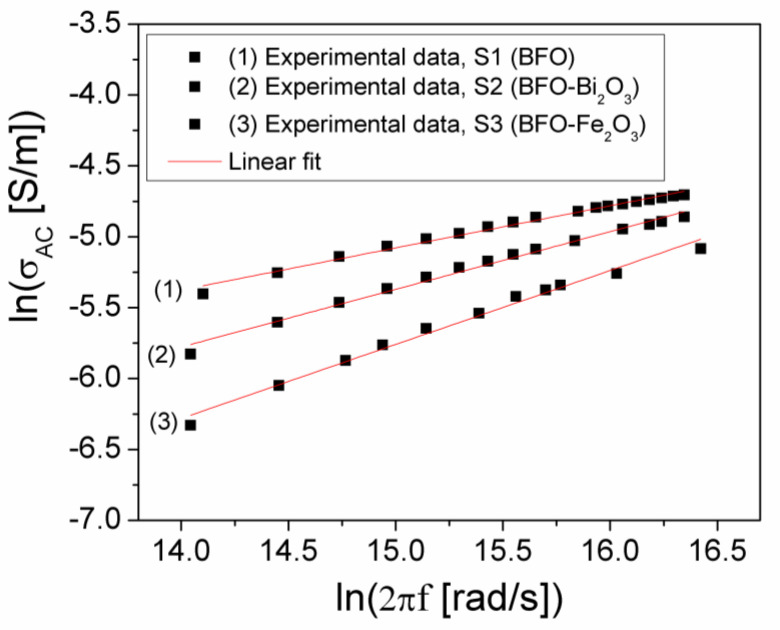
The dependence (*ln*(*σ_AC_*)(*ln*(*ω*)) at room temperature for investigated samples.

**Figure 6 materials-15-04764-f006:**
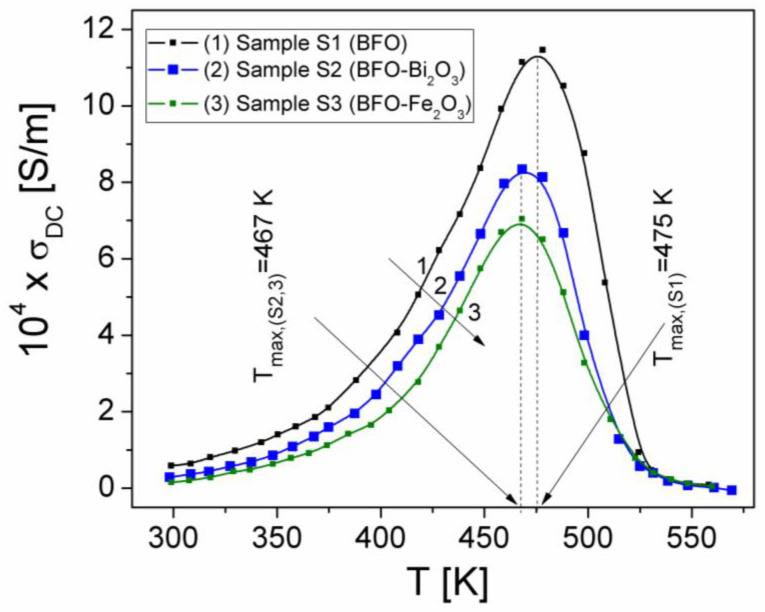
The temperature dependence of the static conductivity (*σ_DC_*) of samples at a constant frequency, f = 0.5 kHz.

**Figure 7 materials-15-04764-f007:**
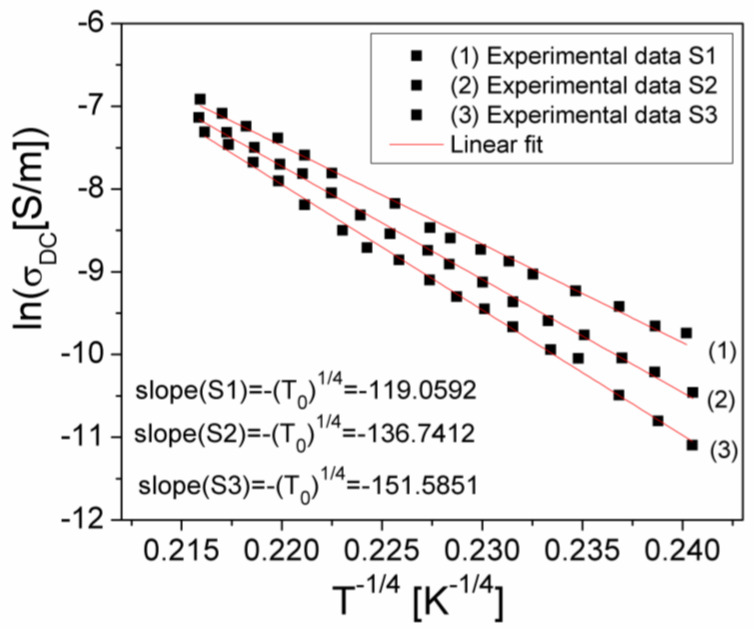
The $\ln\sigma_{DC}(T^{-1/4})$ dependence for investigated samples.

**Figure 8 materials-15-04764-f008:**
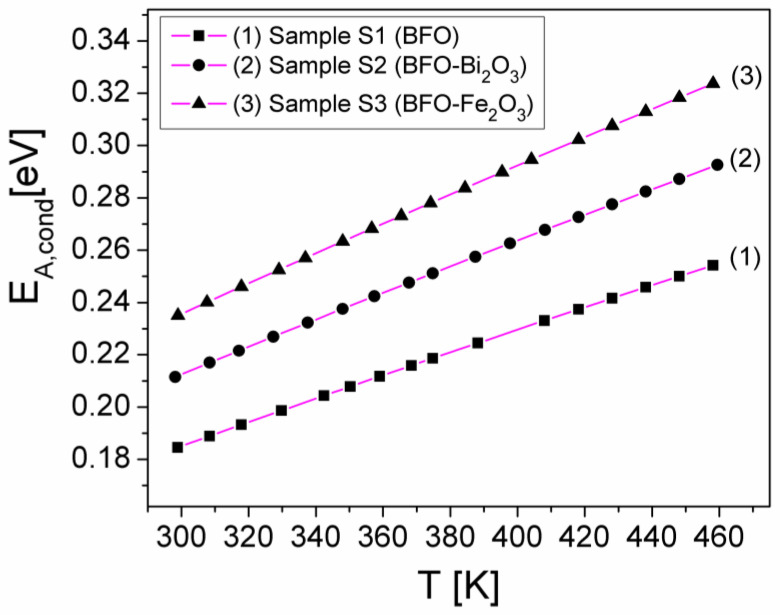
The temperature dependence of the activation energy of the electrical conduction of the investigated samples.

**Figure 9 materials-15-04764-f009:**
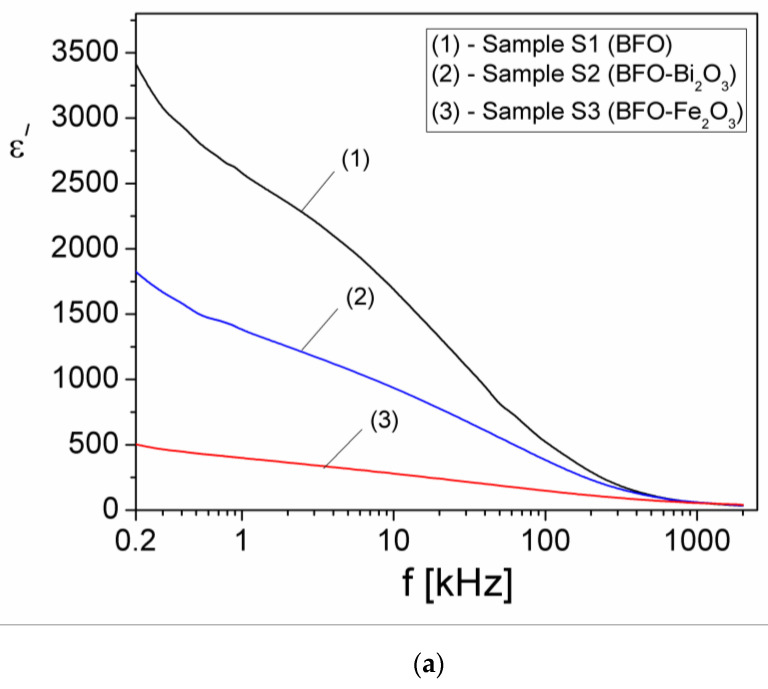
(**a**) The frequency dependence of the real component (ε′) and (**b**) imaginary component ($\varepsilon^{\prime \prime }$ ) of the complex dielectric permittivity of the samples.

**Figure 10 materials-15-04764-f010:**
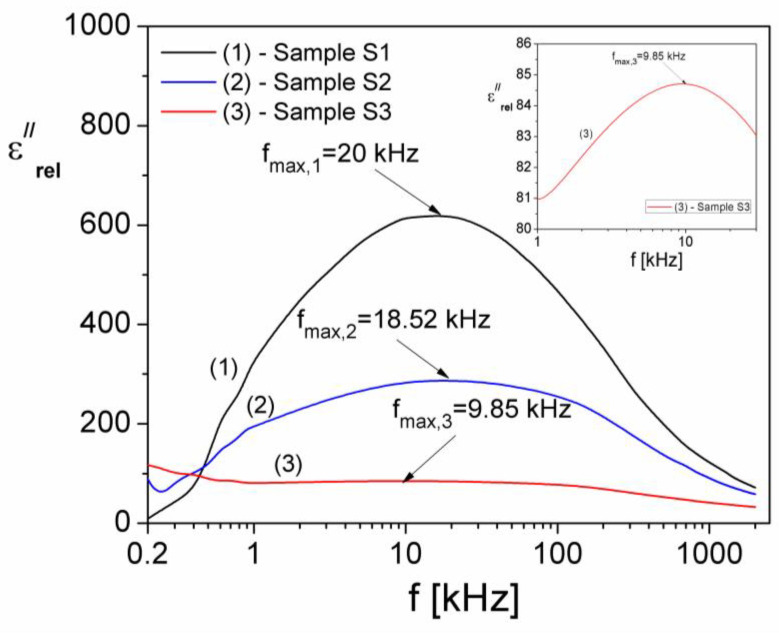
Frequency dependence of the imaginary ($\varepsilon_{rel}^{\prime \prime }$) components due to dielectric relaxation for samples S1, S2 and S3.

**Figure 11 materials-15-04764-f011:**
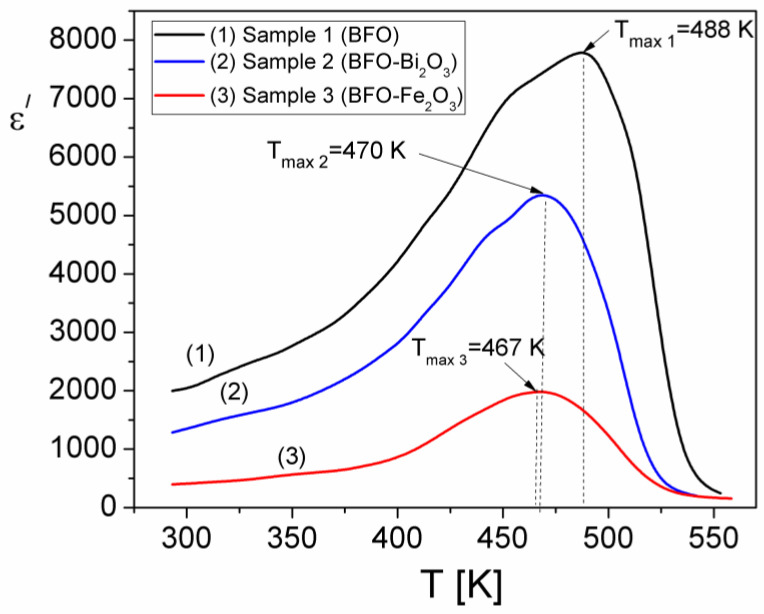
The temperature dependence of the real component (ε′) of complex permittivity for samples at a constant frequency, *f* = 0.5 kHz.

**Table 1 materials-15-04764-t001:** Crystallite size and lattice strain for all samples.

Samples	Crystallite Size (nm)	Lattice Strain (ε)
S1 (BFO)	182.1	0.00198
S2 (BFO-Bi_2_O_3_)	82.5	0.00135
S3 (BFO-Fe_2_O_3_)	62.7	0.000776

**Table 2 materials-15-04764-t002:** The Mott parameters of samples.

Samples	T [K]	N(E_F_) [cm^−3^eV^−1^]	R_h_ [nm]	W_h_ [eV]
S1 (BFO)	318 375 438	6.944 × 10^17^	15.57 14.94 14.37	0.091 0.103 0.116
S2 (BFO-Bi_2_O_3_)	318 375 438	3.446 × 10^17^	18.56 17.80 17.12	0.108 0.123 0.138
S3 (BFO-Fe_2_O_3_)	318 375 438	2.278 × 10^17^	20.57 19.75 18.99	0.120 0.136 0.153

## Data Availability

The data presented in this study are available upon request from the corresponding author.

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
