# Peer review of "The Effect of Bi2O3 and Fe2O3 Impurity Phases in BiFeO3 Perovskite Materials on Some Electrical Properties in the Low-Frequency Field"

_materials, 2022, doi:10.3390/ma15144764_

Round 1

Reviewer 1 Report

Hello, I really enjoyed your manuscript on the conductivity mechanisms of BFO, BFO-Bi2O3, and BFO-Fe2O3. It is a good solid treatment of electroceramics. I have some edits I think are needed to get this manuscript to publishable form.

- On line 53, you have the acronyms SEM and RX. Since this is the first mention of these techniques, please list out the full names first followed by the acronym in parentheses. For example, scanning electron microscope or microscopy (SEM).

- On lines 57 - 60: this sentence is started with "authors have determined some electrical and dielectric properties, but" and then list out some works without stating the contrasting clause. The word "but" is used to separate two contrasting clauses. Either reword or complete this statement.

- On lines 51 - 74: here you define the purpose of your research and your goals. I think the actual purpose is to study the conduction mechanisms of BFO samples and how the impurity phases affects those mechanisms, right? Your research goals or obscured by how you wrote lines 51 - 74. Please consider rewriting to emphasize your goal was to look at the conduction mechanisms.

- For figure 2, the scale bars are really tiny. Please place larger scale bars on the images. Generally speaking, the banner is removed leaving just a scale bar on SEM images.

- For figures 7 and 8, typically the imaginary and real portions of impedance are separated into separate graphs. Please consider separating these to make reading the figures a little easier.

Thank you for your work and good luck.

Reviewer 2 Report

This manuscript reports on the effect of the presence of two common types of impurity phases on the complex impedance (up to 2 MHz) of hydrothermally synthesized BiFeO3. The main conclusion is that the presence of impurity phases decreases the conductivity and modifies related quantities (such as activation energy, DOS at Fermi level) accordingly. Results and analysis are mostly convincing and appear suitable for publication. However, first the authors should address several points listed in the following.

1. The amounts of the impurity phases should be at least roughly quantified, which should be possible from the PXRD data shown in Fig. 1 (even better would be to add a sample with the same type of impurity, but a different amount).

2. The PXRD data of Fig. 1 were probably obtained on the as-grown samples, prior to being pressed into pellets and sintering. Did the authors check with PXRD whether the sintering process involves significant structural changes?

3. The y-axis in the inset of Fig. 3 should probably be labeled as sigma, not sigma_dc: The data also in the inset obviously change with frequency, whereas sigma_dc in the text is described as being frequency-independent. The inset should be explained in the caption also.

4. On the top of page 4, the authors claim that the frequency-dependence of the conductivity follows Jonscher’s law (this is repeated in the conclusions). However, the equation (2) is rather a definition (of sigmaAC, by suctracting from the overall sigma the low-frequency limit) than a law. The point of Jonscher’s law is that the high-frequency part follows a power-law sigmaAC proportional to omega ^ n (with n varying although typically smaller than 1). Such behavior would be checkable by eye in a double-logarithmic plot – but the y-axis in Fig. 3 is linear rather than logarithmic. Power-law behavior should be checked, and n fitted and compared for the three different samples (any temperature-dependence on n would also be of interest).

5. It is confusing that Text and equations use angular frequency omega, but figures use normal frequency f. It may be better to use only one throughout the paper.

6. The observed peak in sigmaDC(T) (and also epsilon’(T) at low frequencies) at 475 K (for the pure sample) should be discussed in the context of previous observations of anomalies in various quantities at similar T in BiFeO3 (see, e.g. the review by Catalan and Scott, Adv. Mater. 21, 2463 (2009), and references therein).

7. Concerning Fig. 5 and Eq. (3), the exact fit-range that was used to extract T0 should be stated.

8. In the presentation of the analysis of the frequency-dependent conductivities (Sec. 3.2.1) and permitivities (Sec. 3.3.1) these quantities are treated as completely different things, whereas they are in fact closely related (including by Kramers-Kronig relations linking real and imaginary parts).

Reviewer 3 Report

The results are properly explained, in general. I have the following recommendations. 

(numbers are line numbers in the right)

25 - "transition in structure", but (crystal) structure is not studied, so electronic structure for example is a better term

37, 42 - proprieties should be properties

41 - "BFO has been intensively studied because of its potential applications in many domains due to both ferroelectric and ferromagnetic proprieties [9]."

Reference 9 does note mention ferroelectric and ferromagnetic as the cause of photocatalytic properties, maybe you can find a better reference here. Also note that BFO has antiferromagnetic properties, not ferromagnetic.

53 - "SEM and RX analysis " SEM and RX abbreviations are not defined. I guess RX is better known as XRD.

148 - Why don't the the sigma_DC values reported in the text for samples 2 and 3 conform with with the values presented in the inset of Fig. 3 for lowest frequencies? 

Fig. 5 T0^1/4 shown in the figure should be positive, due to the minus sign in the equation. But the slope as shown in graph is -T0^1/4. If the authors change T0^1/4 to -T0^1/4 the negative value can be maintained.

Reviewer 4 Report

In  this paper BiFeO3 with Bi2O3 and Fe2O3 impurities have been studied. The study involves microstructural investigation, crystal structure and electrical measurements. The following should be taking into consideration when revising the current manuscript version.

1. The motivation is not clear. The authors state, that addition of some impurity phases into BiFeO3 offers "the possibility to develop a large variety of electronic and electrical devices with new functions". What are these new functions, please name all electronic devices which you have in mind. From the results I only see the decreased conductivity compared to pure compound. I did not see any improvements.

2. The authors state, that at 500 Hz static region of conductivity is obtained. Please add frequency dependencies of the real part of conductivity in all temperatures. Now it is not possible to evaluate the obtained temperature dependences.

3. I have a very similar comment on dielectric permittivity. Have the authors tried to extract epsilon(infinity) parameter from Cole-Cole plots? Even at room temperature eps' does not reach static value at 500 Hz. Why 2 MHz was not considered as being better frequency to represent the dielectric results?

Reviewer 5 Report

In this report, the author’s made single-phase bismuth ferrite (BFO) and BFO with impurity phases (of Bi2O3 and 294 Fe2O3 using the hydrothermal method and studied with extensive measurements techniques and compared the results. My recommendation is minor revision with following comments:

1.       Need to check grammar errors.

2.       What are the % of impurities phases since there are no significant changes with % phases.

3.       Why is the real part of dielectric permittivity increasing with temperature up to some temperature range? Explain?   What kind of transition is expected in conductivity plots? They got a well-defined peak in the conductivity plots.

4.       In the introduction part: some recent references must be added like, Journal of Alloys and Compounds 853, 156979, 2021.

5.       What are the lattice constants for each sample and crystallite size, strain?  They must be also compared. 

Round 2

Reviewer 1 Report

The authors corrected my concerns as well as the concerns of the other reviewers. I feel the manuscript is much improved and ready for publication.